# Model-based stationarity filtering of long-term memory data applied to resting-state blood-oxygen-level-dependent signal

**Ishita Rai Bansal**[1], **Arian Ashourvan**[2,3], **Maxwell Bertolero**[2], **Danielle S. Bassett**[2,3,4,5,6,7], **Sérgio Pequito**[1] *

**1** Delft Centre for Systems and Control, Delft University of Technology, Delft, Netherlands, **2** Department of Bioengineering, School of Engineering and Applied Science, University of Pennsylvania, Philadelphia, PA, United States of America, **3** Penn Center for Neuroengineering and Therapeutics, University of Pennsylvania, Philadelphia, PA, United States of America, **4** Department of Neurology, Hospital of the University of Pennsylvania, Pennsylvania, United States of America, **5** Department of Psychiatry, Perelman School of Medicine, University of Pennsylvania, Philadelphia, PA, United States of America, **6** Department of Electrical & Systems Engineering, School of Engineering and Applied Science, University of Pennsylvania, Philadelphia, PA, United States of America, **7** Department of Physics & Astronomy, College of Arts and Sciences, University of Pennsylvania, Philadelphia, PA, United States of America

* Sergio.Pequito@tudelft.nl

**Data Availability Statement:** The developed code for this work is available in https://github.com/Ishita-Rai/rsfMRI-Fractional-Filter. The data used is available at the Humman Connectome Project

## Abstract

Resting-state blood-oxygen-level-dependent (BOLD) signal acquired through functional magnetic resonance imaging is a proxy of neural activity and a key mechanism for assessing neurological conditions. Therefore, practical tools to filter out artefacts that can compromise the assessment are required. On the one hand, a variety of tailored methods to preprocess the data to deal with identified sources of noise (e.g., head motion, heart beating, and breathing, just to mention a few) are in place. But, on the other hand, there might be unknown sources of unstructured noise present in the data. Therefore, to mitigate the effects of such unstructured noises, we propose a model-based filter that explores the statistical properties of the underlying signal (i.e., long-term memory). Specifically, we consider autoregressive fractional integrative process filters. Remarkably, we provide evidence that such processes can model the signals at different regions of interest to attain stationarity. Furthermore, we use a principled analysis where a ground-truth signal with statistical properties similar to the BOLD signal under the injection of noise is retrieved using the proposed filters. Next, we considered preprocessed (i.e., the identified sources of noise removed) resting-state BOLD data of 98 subjects from the Human Connectome Project. Our results demonstrate that the proposed filters decrease the power in the higher frequencies. However, unlike the low-pass filters, the proposed filters do not remove all high-frequency information, instead they preserve process-related higher frequency information. Additionally, we considered four different metrics (power spectrum, functional connectivity using the Pearson's correlation, coherence, and eigenbrains) to infer the impact of such filter. We provided evidence that whereas the first three keep most of the features of interest from a neuroscience perspective unchanged, the latter exhibits some variations that could be due to the sporadic activity filtered out.

website, and its preprocessing methods are described within the main manuscript and references therein. The data of the Humman Connectome Project can be accessed by all at http://www.humanconnectomeproject.org/.

**Funding:** D.B. work is supported by the National Institutes of Health grants 5-T32-NS-091006-07, 1R01NS116504, 1R01NS099348, 1R01NS085211, and 1R01MH112847. D.B. also acknowledge support by the Thornton Foundation, the Mirowski Family Foundation, the ISI Foundation, the John D. and Catherine T. MacArthur Foundation, the Sloan Foundation, the Pennsylvania Tobacco Fund, and the Paul Allen Foundation. SP was partially supported by NSF CMMI 1936578.

**Competing interests:** The authors have declared that no competing interests exist.

## Introduction

Functional magnetic resonance imaging (fMRI) has gained popularity as a noninvasive method for measuring brain activity across different brain regions. The fMRI studies assess the fluctuations in blood oxygenation level-dependent (BOLD) signals of the brain generated as a time series during either rest or as a response to some task or externally applied stimulus. The observations by Biswal *et al.* about the temporal correlation between the BOLD fluctuations of left and right hemispheric regions of the primary motor cortex during rest laid the foundation for a new era of research for understanding the neuroanatomy by analysing the resting-state fMRI (rs-fMRI) time series data [1]. From its discovery, a rapid growth in rs-fMRI literature has been witnessed [2] in order to understand the state of brain in neurodegenerative disease, psychological behaviour, effect of anaesthesia, among other applications [3–8].

The fluctuations in BOLD signal in rs-fMRI studies consists of contribution from neural activity of the brain and also from several physiological and non-physiological sources. Physiological sources of noise include artefacts arising from both neuronal and non-neuronal components such as due to cardiac and respiratory cycles, blood pressure oscillations [9, 10]. Non-physiological noises include fluctuation due to drift, slice time correction, subject motion and hardware instabilities [9, 11, 12]. Marcus *et al.* in [13] showed that of the total variance in resting-state BOLD data considered by them, nuisance components accounted for 16% of variance, motion regressors for 14%, neural components for 4% and rest was due to other unstructured artefacts. These confounds pose a risk of artificially influencing the functional connectivity between different brain regions and thus yielding spurious results [10]. The composition of the BOLD signal and the examples mentioned highlight the fact that the study of FC of the brain by resting-state BOLD is highly data-driven and hence emphasise the importance of preprocessing of rs-fMRI data for artefact removal.

Many denoising approaches have been developed for isolating true neural activity from acquired resting-state BOLD data. These include the following: *(i)* model-based approaches which estimate contributions to BOLD signal from physiological sources [10, 14] or due to head motion [15]; *(ii)* data-driven approaches which estimate noise from data using ICA [11]; *(iii)* scrubbing (removing) time points acquired during period of high motion [16–18] and *(iv)* combining data-driven methods with multiecho data acquisitions, which were observed to perform better in terms of removing noise from BOLD signal fluctuations [12, 19].

Nonetheless, even after the application of different preprocessing methods, the rs-fMRI data obtained may include signal fluctuations due to unknown unstructured sources of noise or spurious fluctuations caused by the reintroduction of some artefacts previously removed in preprocessing steps in later preprocessing steps [20]. In what follows, we propose to take preprocessed (i.e., free from motion and nuisance artefacts) resting-state BOLD signals and perform time-domain filtering using a parametric filter to mitigate the effects of unstructured noise due to unknown sources.

The presence of long-term memory in the data as captured by their autocorrelation functions led us to propose a univariate autoregressive fractional integral moving average (ARFIMA) [21] model-based filter that is suitable to retain the signal with long-term memory and filter out the signal (noise) which does not conform with the long-term autocorrelation structure nor with the white Gaussian noise driving the process. In fact, this is the first time such a filter is proposed for this purpose to the best of the authors' knowledge. In the context of this study, whenever we refer to the resting-state BOLD dataset, we mean the preprocessed signals and filtered BOLD signals, imply, the signals obtained after ARFIMA filtering.

The results hereafter suggest that such processes are suitable to model resting-state BOLD. The proposed filter in addition to attenuating higher frequencies, is able to preserve process-related information in these frequencies.

## Materials and methods

### Dataset and preprocessing

We considered rs-fMRI BOLD dataset from the Human Connectome Project (HCP) [22] that was preprocessed by us. Specifically, the resting-state BOLD data was acquired in two sessions consisting of two runs each of approximately 15 minutes. Within each session, in one run, phase encoding was done in the right-to-left (RL) direction and left-to-right (LR) in another run. Participants were instructed to relax with their eyes open and visual fixation on a projected bright cross-hair on a dark background (and in a darkened room).

The data was acquired on HCP 3T Siemens "Connectome Skyra" scanner and was part of the HCP S1200 release. The rs-fMRI data was obtained using BOLD contrast sensitive gradient-echo echo-planar imaging having a multiband factor of 8, TE of 33.1 ms, TR of 720 ms, a spatial resolution of 2 mm isotropic voxels and a flip angle of 52 deg ([22, 23] provides the detailed description of acquisition protocol). The HCP rs-fMRI dataset was first preprocessed by the Human Connectome Project using FMRIB's ICA-based Xnoiseifier (ICA-FIX) methodology [22, 24, 25] to remove artefacts related to motion and nuisance signals. In the next step, we applied global signal regression preprocessing [26], and the obtained preprocessed data is used for further analysis in the research. We considered the dataset of 98 subjects with the least head movement artefacts. The dense time series of each subject is cortically parcellated [27] into $n(= 100)$ brain regions of interest (ROI). The BOLD signal is collected from each ROI at 1200 time points for 4 different runs for each subject. Therefore, in what follows, we consider a total of $n(= 100)$ time series with 1200 data points of the resting-state BOLD signal for each of the four runs of each subject.

### ARFIMA modelling

Let us consider a stationary stochastic process $X_t$, $t \in \mathbb{N}$, whose time series realization $x_t$, $t \in 1, \ldots, M$, consists of $M$ successive observations made at equidistant time intervals. In the context of our study this will represent the resting-state BOLD data of a single region of interest. We can use autoregressive fractional integral moving average (ARFIMA) model to model the stochastic process $X_t$ [28–30]. Therefore, we obtain an ARFIMA $(p, d, q)$ model, where the parameters $p$, $d$ and $q$ describe the order of the autoregressive, fractional integrative, and moving average component respectively. Specifically, an ARFIMA $(p, d, q)$ is described by:

$$\phi(B)(1 - B)^d (X_t - \mu) = \psi(B)\varepsilon_t, \tag{1}$$

where $\mu$ is the mean of the process $X_t$, $\varepsilon_t$ is the independent and identically distributed (i.i.d.) noise with mean zero and bounded variance. In particular, we can assume the noise to be white gaussian noise, which is described by a normal distribution with zero mean and variance $\sigma_\varepsilon^2$, denoted by $\varepsilon_t \sim WN(0, \sigma_\varepsilon^2)$. Additionally, $(1 - B)^d$ is the difference operator, $d \in \mathbb{R}_+$ is the *fractional-order* difference parameter, $B$ is the backshift operator [31]. Furthermore, $\phi(B)$ and $\psi(B)$ are the autoregressive and moving average operators, respectively. The autoregressive and moving average operators represented in terms of backshift operator are defined as:

$$\phi(B) = 1 - B\phi_1 - B^2\phi_2 - \ldots - B^p\phi_p \tag{2}$$

and

$$\psi(B) = 1 + B\psi_1 + B^2\psi_2 + \ldots + B^q\psi_q, \tag{3}$$

where $\phi_1, \phi_2, \ldots, \phi_p \in \mathbb{R}$ are the autoregressive parameters, $p \in \mathbb{N}$ is the order of the autoregressive component, $\psi_1, \psi_2, \ldots, \psi_q \in \mathbb{R}$ are the moving average parameters and $q \in \mathbb{N}$ is the order of moving average component.

Differencing operation is used to remove non-stationarity and long-term memory property present in the time series. The difference operator can be represented in terms of the gamma function by solving its binomial expansion when $d > -1$ [32] as

$$(1 - B)^d = \sum_{k=0}^{\infty} \frac{\Gamma(k - d)}{\Gamma(k + 1)\Gamma(-d)} B^k, \tag{4}$$

where $\Gamma(\cdot)$ is the gamma function defined by $\Gamma(z) = \int_0^{\infty} x^{z-1}e^{-x}dx$, for all complex numbers with $\Re(z) > 0$ [33]. Notwithstanding, we can generalize for any value of $d$ by noticing that the operator $(1 - B)$ is linear. Specifically, for any $d' \in \mathbb{Z}$ and $d > -1$, we can have $(1 - B)^{d'+d} = (1 - B)^{d'}(1 - B)^d$, where the first term is a difference equation and the second is given in Eq 4.

Subsequently, we performed the fractional differencing of the time series as follows:

$$
\begin{aligned}
(1 - B)^d x_t \quad &= \left( \sum_{k=0}^{\infty} \frac{\Gamma(k - d)}{\Gamma(k + 1)\Gamma(-d)} B^k \right) x_t, \\
&= (1 - \mathrm{d}B + \mathrm{d}(d - 1)B^2/2! - \ldots)x_t \\
&= x_t - dx_{t-1} + d(d - 1)x_{t-2}/2! - \ldots .
\end{aligned}
\tag{5}
$$

Notice that the fractional differencing operation has an infinite impulse response (as can be seen from the infinite series expansion of fractional differencing); specifically, it models an infinite order autoregressive process whose parameters are defined by the fractional differencing weights. For the practical implementation, this infinite impulse response process is converted to finite impulse response by limiting the number of weights (i.e., by truncating the series). Specifically, we only consider weights whose absolute value is greater than 0.0001.

**ARFIMA filtering on resting-state BOLD signals.** We perform ARFIMA $(p, d, q)$ (model-based) filtering on each resting-state BOLD time series. The procedure of ARFIMA $(p, d, q)$ filtering can be broadly divided into two parts: *(i)* estimation of the fractional difference parameter followed by determining the autoregressive and moving average parameters of the filter, and *(ii)* use the ARFIMA model to obtain an infinite impulse response filter which is used to filter out any potential noise present in the data, see flowchart in Fig 1.

First the fractional difference parameter $d$ in Eq 1 is estimated from resting-state BOLD data using its sampled autocorrelation function (sACF). Specifically, resting-state BOLD time series is fractionally differenced for different values of $d$ in the range [0.1, 5.0] and the number of statistically significant lags (i.e., the lags for which autocorrelation value lies outside the range $\pm 2/\sqrt{N}$) are calculated. The final value of $d$ considered is that for which the number of statistically significant lags is minimum.

The ARMA models assume that the process is stationary [30]. Therefore, we inspect that the differentiated resting-state BOLD data with the selected parameter $d$ satisfies the Kwiatowksi, Phillips, Schmidt and Shin (KPSS) test [34]. This statistical test tests the null hypothesis

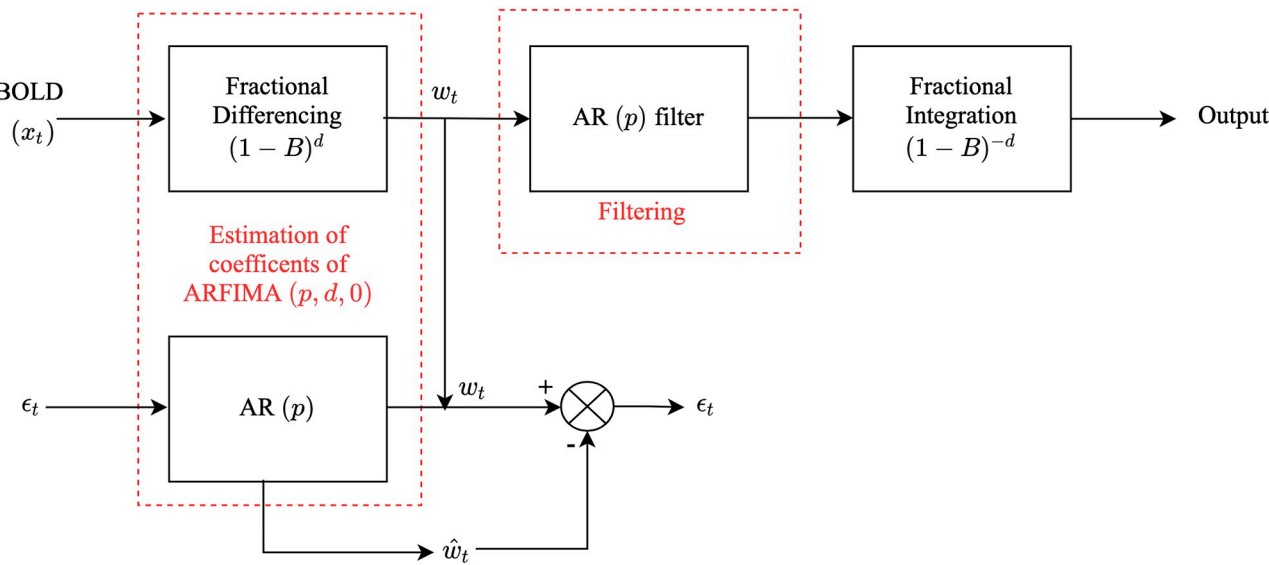

**Fig 1. Block diagram of the proposed method to filter resting-state BOLD data.**

that the observed time series is stationary around a deterministic trend (for a significance level of $p < 0.05$).

In the next step, we process towards the estimation of the parameters $p$ and $q$ of ARFIMA $(p, d, q)$ process. In order to devise a causal filter [30], the moving average parameter $q$ is set to 0. Therefore, in this study, we do not use the moving average component, which, reduces ARFIMA model to ARFI (Autoregressive fractional integrative) model. Since the commonly used term is ARFIMA modelling, therefore, we use the same terminology in this study.

Due to the inherent existence of an infinite order autoregressive process in the fractional differencing operation in ARFIMA $(p, d, q)$ modelling, we limit the order $p$ to 1. This helps in restricting the number of degrees of freedom in the process. The infinite autoregressive parameters, in this case, are specified by one parameter, $d$.

Additionally, in order to check that indeed AR (1) process is capable of capturing the dynamics of the fractionally differenced stationary time series, $w_t$, we start by fitting an AR model of order 1. The coefficient ($\phi$) of this model is estimated using maximum likelihood principle [30] and the residual between $w_t$ and the simulated time series $\hat{w}_t$ from the fitted model are observed. If the residual error is statistically indistinguishable from white noise (i.e., having flat power spectrum and uncorrelated residuals), then it suggests that there is no need for higher-order fitting; otherwise order $p$ is increased, and then residuals are tested again [35, 36]. We performed student's $t$-test [37] to test the statistical significance that the residual error follows a normal distribution (at a significance level of $p < 0.05$). In our case, the residuals behaved as a standard white noise process for an AR process of order 1.

Finally, after the estimation of the AR (1) coefficient, the resting-state BOLD data is filtered through the designed infinite impulse response AR (1) filter.

The above procedure is repeated for each of the resting-state BOLD data from $n(= 100)$ ROIs of all 98 subjects in all 4 runs to obtain ($100 \times 98 \times 4 = 39200$) ARFIMA (1, $d$, 0) filtered resting-state BOLD time series. In the procedure, the value of fractional difference parameter $d$ and AR coefficient $\phi$ varies for each BOLD time series. Fig 2 shows the mean of the estimated values of the AR (1) parameter, $\phi$ across different ROIs. Lastly, Table 1 shows the mean value of $d$ for $n(= 100)$ ROI averaged across all subjects in all runs. Thus, implying that this ARFIMA

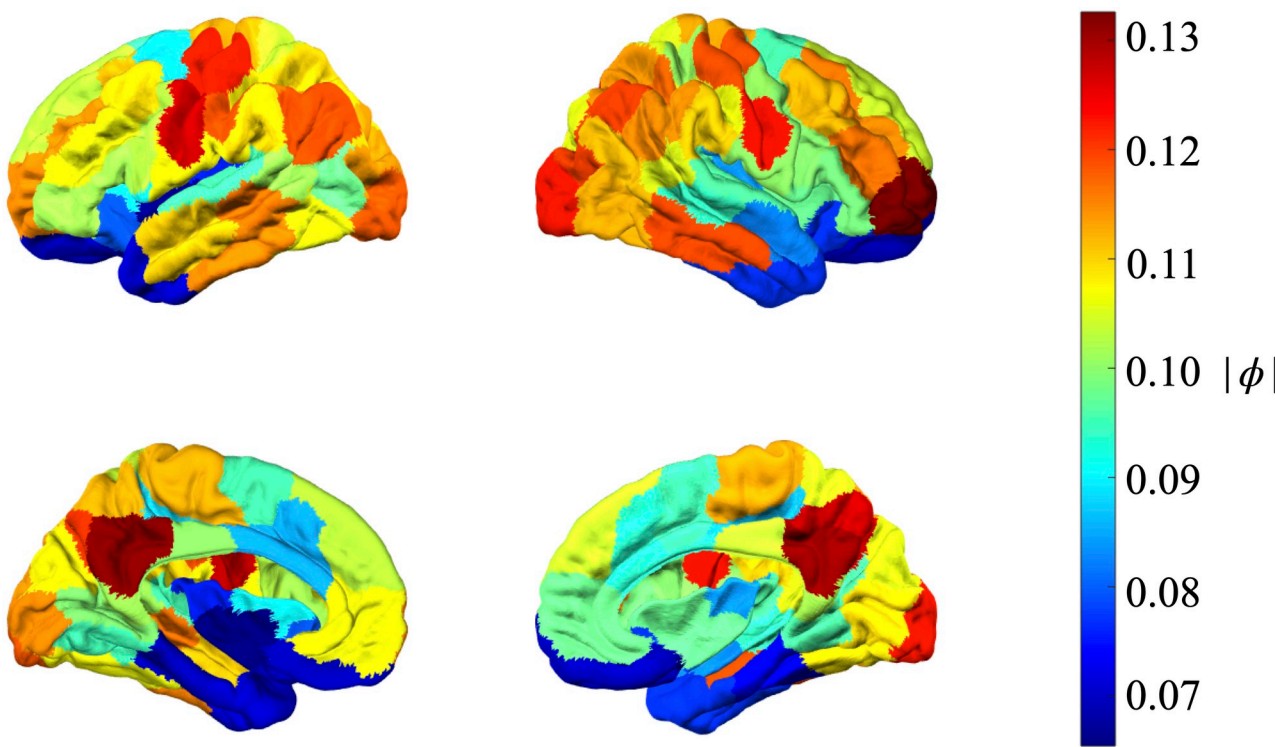

**Fig 2. Mean value of the absolute AR(1) parameter averaged across all 98 subjects across all 4 runs.**

$(1, d, 0)$ filtering approach is deployed channel-wise. Additionally, Kruskal-Wallis test [38] on both parameters $\phi$ and $d$ showed that the mean between these parameters for different subjects across different runs was significantly different (at a significance level of 0.05).

## Results

This section presents the results of the proposed model-based ARFIMA filtering on a synthetic example and the resting-state BOLD time series based upon different measures: normalised power spectrum, functional connectivity (FC) (i.e., Pearson's correlation and coherence) and eigenmode analysis of directed connectivity.

Specifically, we consider a synthetic BOLD signal which has properties similar to the resting-state BOLD signal (e.g., long-term memory [39–41] and higher power in lower frequencies [9, 42, 43] as captured by the sACF and power spectrum, respectively), to which white noise is added. Afterwards, the designed fractional filter is applied to the original resting-state BOLD data.

### Synthetic BOLD signal

To validate the impact of the proposed methodology to filter out the effect of unwanted fluctuations or fluctuations due to unstructured noise, we created a synthetic signal such that we have a "ground-truth" signal. We generated a synthetic BOLD signal as a sum of sinusoidal signal of different frequencies to obtain a signal with long-term memory property and specific power spectrum, mathematically represented as

$$x_t = \sum_{i=1}^{N} A_i \sin(2\pi f_i t), \tag{6}$$

**Table 1. Mean value of the fractional difference parameter *d* averaged across the ROIs of all 98 subjects in all 4 sessions (i.e., 98 × 4).**

| ROI | d (mean ± std. deviation) | ROI | d (mean ± std. deviation) | ROI | d (mean ± std. deviation) | ROI | d (mean ± std. deviation) |
|---|---|---|---|---|---|---|---|
| 1 | 0.6679 ± 0.1779 | 26 | 0.6566 ± 0.1636 | 51 | 0.6051 ± 0.1860 | 76 | 0.6849 ± 0.1396 |
| 2 | 0.9617 ± 0.1751 | 27 | 0.3702 ± 0.1398 | 52 | 0.9411 ± 0.1770 | 77 | 0.4186 ± 0.1269 |
| 3 | 0.7202 ± 0.2031 | 28 | 0.1946 ± 0.1036 | 53 | 0.9130 ± 0.1762 | 78 | 0.1880 ± 0.0980 |
| 4 | 0.9416 ± 0.1769 | 29 | 0.2464 ± 0.1112 | 54 | 0.7268 ± 0.2078 | 79 | 0.2292 ± 0.1200 |
| 5 | 0.5003 ± 0.1811 | 30 | 0.4533 ± 0.1769 | 55 | 0.4906 ± 0.1643 | 80 | 0.7051 ± 0.1501 |
| 6 | 0.5191 ± 0.1796 | 31 | 0.8339 ± 0.1467 | 56 | 0.8758 ± 0.1753 | 81 | 0.6849 ± 0.1500 |
| 7 | 0.7643 ± 0.1936 | 32 | 0.7163 ± 0.1451 | 57 | 0.4406 ± 0.1344 | 82 | 0.6003 ± 0.1437 |
| 8 | 0.7209 ± 0.1926 | 33 | 0.6227 ± 0.1436 | 58 | 0.6865 ± 0.1809 | 83 | 0.6967 ± 0.1691 |
| 9 | 0.6110 ± 0.1782 | 34 | 0.5980 ± 0.1502 | 59 | 0.4722 ± 0.1642 | 84 | 0.9449 ± 0.1646 |
| 10 | 0.5097 ± 0.1778 | 35 | 0.7921 ± 0.1466 | 60 | 0.6158 ± 0.1790 | 85 | 0.7056 ± 0.1501 |
| 11 | 0.2804 ± 0.1293 | 36 | 0.6031 ± 0.1508 | 61 | 0.4934 ± 0.1772 | 86 | 0.7140 ± 0.1655 |
| 12 | 0.5640 ± 0.1562 | 37 | 0.3344 ± 0.1371 | 62 | 0.3161 ± 0.1343 | 87 | 0.3158 ± 0.1285 |
| 13 | 0.7452 ± 0.1917 | 38 | 0.6421 ± 0.1441 | 63 | 0.5069 ± 0.1516 | 88 | 0.7212 ± 0.1320 |
| 14 | 0.6209 ± 0.1615 | 39 | 0.7008 ± 0.1496 | 64 | 0.7457 ± 0.1732 | 89 | 0.8622 ± 0.1764 |
| 15 | 0.6370 ± 0.1719 | 40 | 0.6306 ± 0.1481 | 65 | 0.7110 ± 0.1711 | 90 | 0.6770 ± 0.1477 |
| 16 | 0.7791 ± 0.1689 | 41 | 0.6306 ± 0.1716 | 66 | 0.6541 ± 0.1658 | 91 | 0.6599 ± 0.1425 |
| 17 | 0.6969 ± 0.1779 | 42 | 0.6719 ± 0.1578 | 67 | 0.7821 ± 0.1678 | 92 | 0.5258 ± 0.1444 |
| 18 | 0.5747 ± 0.1651 | 43 | 0.9824 ± 0.1789 | 68 | 0.6648 ± 0.1723 | 93 | 0.7344 ± 0.1601 |
| 19 | 0.5714 ± 0.1709 | 44 | 0.8013 ± 0.1789 | 69 | 0.6084 ± 0.1615 | 94 | 0.3426 ± 0.1427 |
| 20 | 0.5347 ± 0.1485 | 45 | 0.6819 ± 0.1537 | 70 | 0.5227 ± 0.1531 | 95 | 0.6395 ± 0.1733 |
| 21 | 0.6875 ± 0.1536 | 46 | 0.3482 ± 0.1529 | 71 | 0.7712 ± 0.1670 | 96 | 0.4566 ± 0.1344 |
| 22 | 0.2370 ± 0.1408 | 47 | 0.7358 ± 0.1775 | 72 | 0.4531 ± 0.1301 | 97 | 0.2566 ± 0.1188 |
| 23 | 0.4365 ± 0.1363 | 48 | 0.4526 ± 0.1259 | 73 | 0.3495 ± 0.1186 | 98 | 0.3880 ± 0.1505 |
| 24 | 0.3495 ± 0.1317 | 49 | 0.2503 ± 0.1807 | 74 | 0.4622 ± 0.1551 | 99 | 0.5635 ± 0.1582 |
| 25 | 0.4561 ± 0.1528 | 50 | 0.5658 ± 0.1637 | 75 | 0.8531 ± 0.1610 | 100 | 0.5685 ± 0.1727 |

where $x_t$ is the synthetic signal, $A_i$ and $f_i$ is the amplitude and frequency of the i*th* signal, respectively, and *N* is the number of signals. For the purpose, we generated a random vector of 10 different frequencies in the range $0.1 - 0.15$ Hz to mimic the presence of the low-frequency fluctuations in the resting-state BOLD signals.

The generated BOLD signals have properties similar to the original resting-state BOLD signal (from the dataset). Specifically, the synthetic signal was sampled at 1.3889 Hz in order to emulate the sampling frequency of the resting-state BOLD signal. Fig 3 compares the time-series and power spectrum plot of the original rs-BOLD signal and generated synthetic BOLD. Here, we present the results from one of the synthetic BOLD signal. The long-term memory property (which can be seen from inverse power law envelop on the sACF plot on the left-hand side in Fig 4A) and fluctuations in the low-frequency range $0.1 - 0.15$ Hz (plot on the right-hand side in Fig 4A) can be observed in the generated signal. It was then corrupted with two different white gaussian noise sequence, $\varepsilon_t \sim WN(0, 100)$ and $WN(0, 10)$. After the addition of noise, we use the proposed methodology to implement ARFIMA filtering to get a filtered synthetic BOLD signal. The parameters of the ARFIMA filter for the shown synthetic BOLD signal with variance of noise 100 and 10 were identified as ARFIMA (1, 2.6, 0) and ARFIMA (1, 3.8, 0), respectively. Fig 4B and 4C provides the sACF and normalized power spectrum of the filtered synthetic BOLD signal in both the cases. Similar results were obtained with other synthetic BOLD signals.

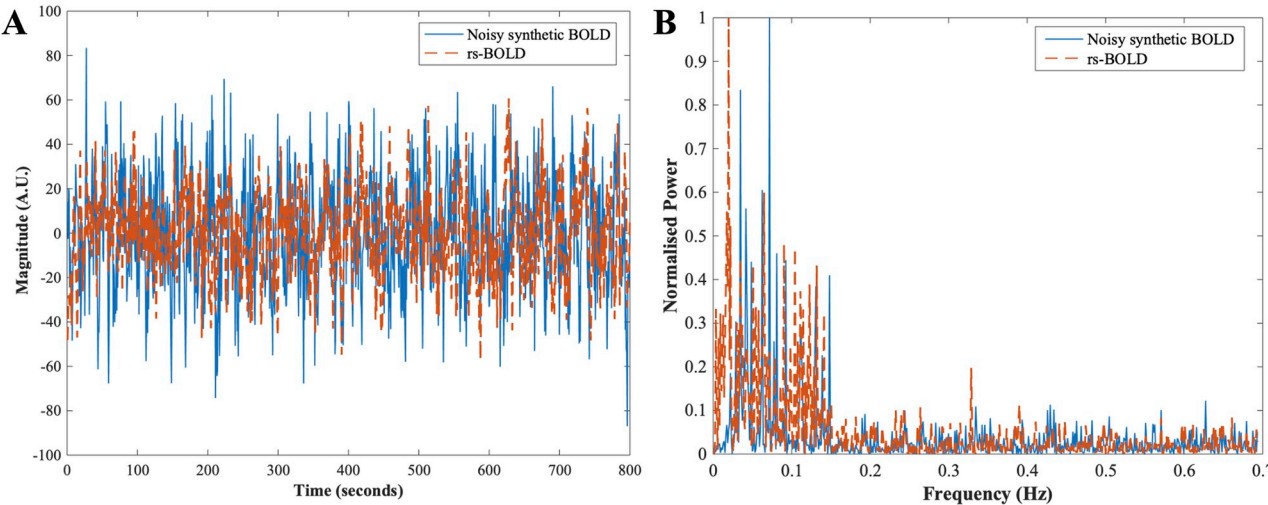

**Fig 3. Comparison between the generated noisy synthetic BOLD and the rs-BOLD signal from one of the ROI from dataset.** (A) The time series plot of the generated noisy synthetic BOLD (cyan-colored) and preprocessed rs-BOLD (dashed orange curve). (B) Power spectrum plot of the noisy synthetic BOLD (cyan) and rs-BOLD (orange).

## Resting-state BOLD signal

We perform both subject level and group level analysis to show the effect of proposed ARFIMA filtering on the resting-state BOLD data. Notably, as highlighted in Table 1, the value of parameter $d$ varies for each of the ROI for each subject and also across runs.

**Normalised power spectrum of the ARFIMA filtered resting-state BOLD data.** In order to show the spatial disparity in the power spectrum, we illustrate the results of the proposed filtering on the power spectrum of three different ROIs highlighted in red/green/blue on the brain surface of one of the subject in Fig 5. The three shown power spectrums are such that they have different characteristics in terms of the presence of power in the frequency region. Specifically, Fig 5A depicts the effect of filtering on the normalised power spectrum of ROI: 7 (present in the visual peripheral network of the brain), which has maximum power in the lower frequencies. In contrast, Fig 5B shows the effect of filtering on the normalised power spectrum of ROI: 11 which lies in the somatomotor network (has power spreaded out in whole frequency range) and Fig 5C corresponds to the ROI: 37 lying in the executive control network (composed of the maximum power in lower frequencies but significant amount of power in higher frequency region).

We assessed the statistical similarity of the normalised power spectrum of each ROI before and after filtering, using two-sample Kolmogorov–Smirnov test [44] at a significance level of 0.05. The statistical comparison between the normalised power spectrum of each of the ROI of all the subjects across all runs before and after proposed filtering indicates that of all the power spectrum corresponding to total $98 \times 4 \times 100 = 39200$ resting-state BOLD signals, around 51% (i.e., 19966 signals) of the them were statistically distinguishable. Fig 6 shows the total number of BOLD signals whose power spectrum were statistically different in each ROI.

**Functional connectivity measures: Pearson's correlation and coherence of the ARFIMA filtered resting-state BOLD data.** Functional connectivity (FC) is defined as the temporal co-activation in the measured brain signals between two ROIs. The FC matrix is a $n \times n$ symmetrical matrix, where $n$ is the number of ROIs in which the brain is parcellated. Each element of the FC matrix defines the strength of the connection between two ROIs. We display the

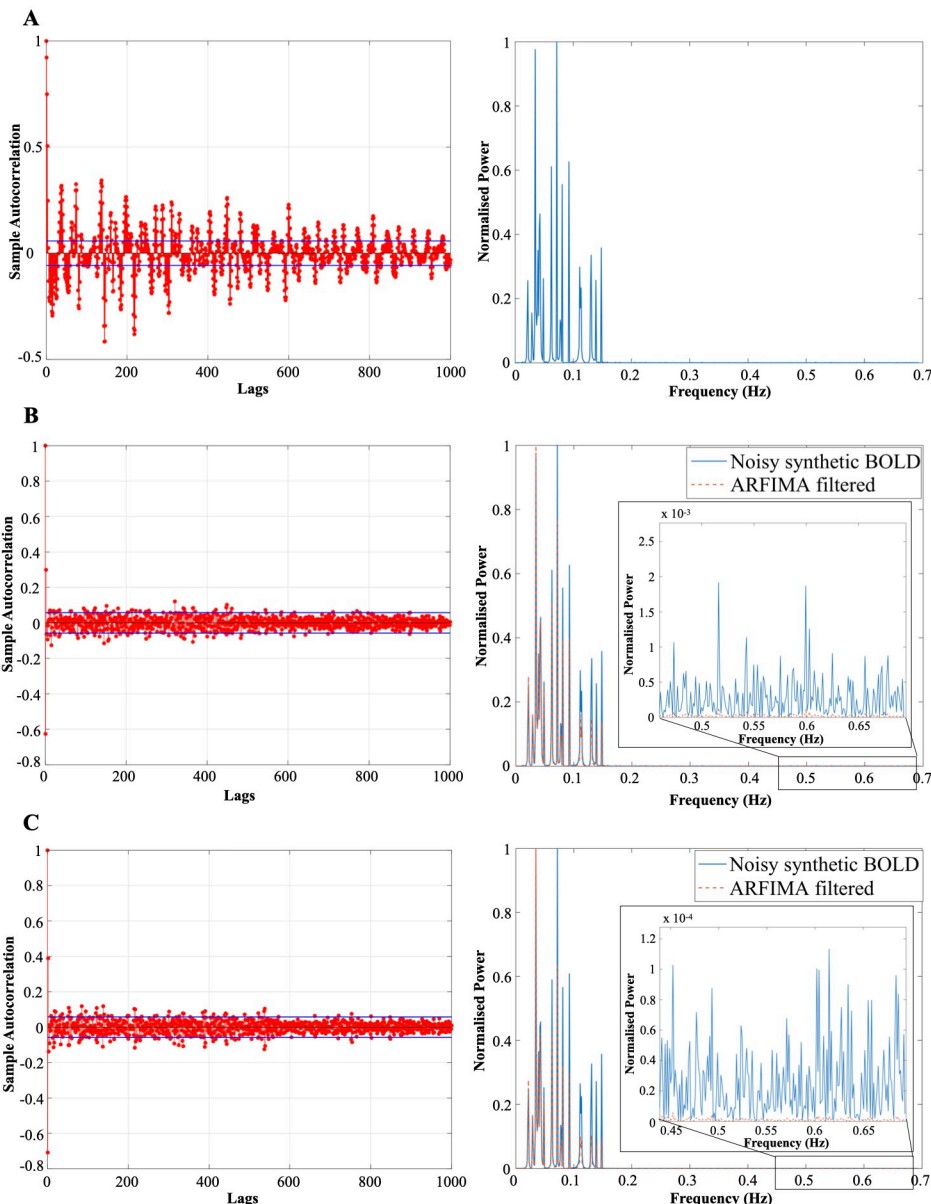

**Fig 4. Autocorrelation and normalised power spectrum plots of unfiltered and filtered synthetic BOLD signal.** (A) The designed synthetic BOLD signal with a sampling frequency of 1.3889 Hz (similar to the sampling rate of rs-fMRI HCP dataset) which has inverse power law autocorrelation plot (left panel) (similar to the observed original resting-state BOLD signal). The synthetic BOLD signal thus created consists of low frequency fluctuations in the range of 0.01 − 0.15 Hz (similar to the observation about intrinsic BOLD fluctuation in the resting brain in [9, 42, 43]), right panel. An artificial white noise is added to the created synthetic BOLD signal. (B) The left panel shows the sACF of the fractionally differenced ($d$ = 2.6) synthetic BOLD signal (with white noise, $WN(0, 100)$). The right panel shows the normalised power spectrum plot of unfiltered dummy BOLD signal (cyan curve, $WN(0, 100)$) and the ARFIMA (1, 2.6, 0) filtered BOLD signal (dashed orange curved) embedded with zoomed in plot at higher frequency. (C) The sACF of the fractionally differenced ($d$ = 3.8) synthetic BOLD signal (with white noise, $WN(0, 10)$) and normalised power spectrum plot of unfiltered dummy BOLD signal (cyan colored curve, $WN(0, 10)$) and the ARFIMA (1, 3.8, 0) filtered BOLD signal (dashed orange curve) embedded with zoomed in plot at higher frequency is depicted in the left and right panel respectively.

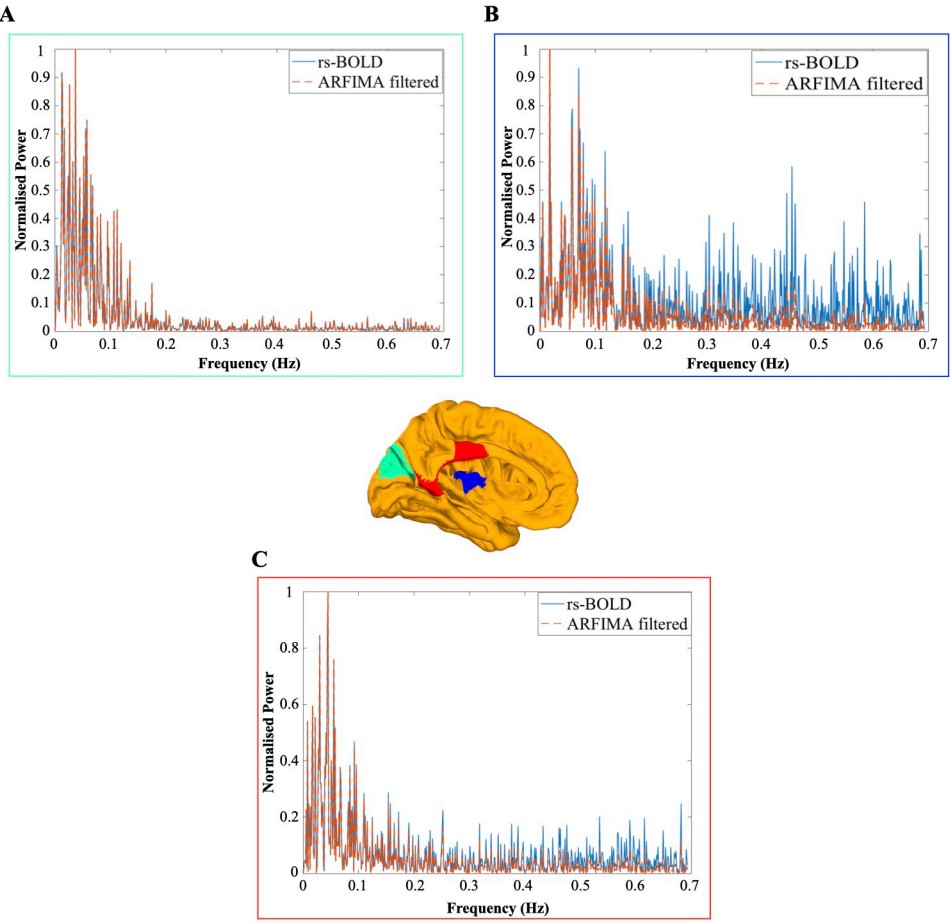

**Fig 5. Normalised power spectrum of the unfiltered and filtered resting-state BOLD signal for three different ROIs.** The color-coded power spectrum plots, i.e. cyan and dashed orange, represent the plots of the resting-state BOLD signal and ARFIMA (1, *d*, 0) filtered BOLD signal respectively of one of the subjects. The location of each of the three ROI is presented in the brain overlay (in the centre) in different colours. The power spectrum of the corresponding ROI is outlined in the same colored box. (A-C) Normalised power spectrum of the resting-state and ARFIMA filtered BOLD signal corresponding to the ROI: 7 lying in the visual peripheral brain network (a green colored region in brain), ROI: 11 in the somatomotor auditory network of the brain (a blue colored region in the brain) and ROI: 37 in control network (a red colored region), respectively, is shown. The filter used in each of the case is: ARFIMA (1, 0.7, 0) (in A), ARFIMA (1, 0.3, 0) (in B) and ARFIMA (1, 0.5, 0) (in C).

results of two different FC measures (i.e., Pearson's correlation and coherence) of the resting-state BOLD and ARFIMA filtered resting-state BOLD in Figs 7 and 8. Specifically, Fig 7 illustrates the results of group-level analysis of the Pearson's correlation FC, whereas Fig 8 depicts the FC matrix using the coherence. For an overview on Pearson Correlation and coherence, see S1 and S2 Sections in S1 File. Lastly, we emphasise that the two-sample Kolmogorov–Smirnov statistical test [44] between the mean FC matrices of preprocessed and filtered BOLD signals fails to reject the null hypothesis that the FC matrices are statistically indistinguishable ($p < 0.05$).

**Eigenmode analysis of directed connectivity: Before and after proposed filtering.** Directed functional connectivity is defined as the system that one obtains from the linear time-invariant approximation of the underlying dynamics of the BOLD signals. The system, thus, obtained can be decomposed into its so-called eigenmodes. Eigenmodes (represented by

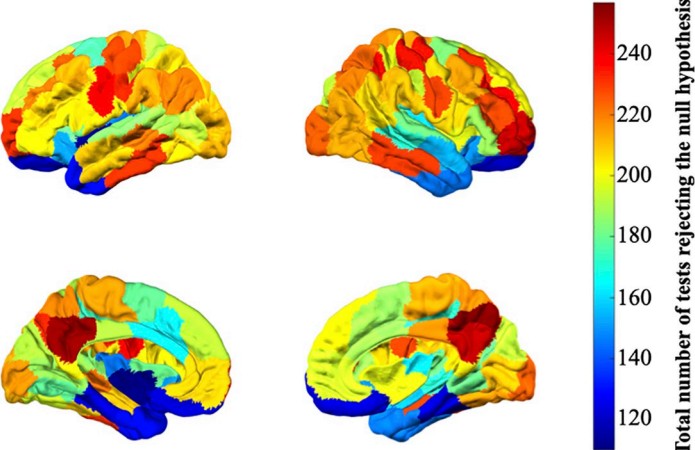

**Fig 6. Number of statistically distinguishable power spectrum ($p < 0.05$) in each ROI plotted on the brain overlays.**

corresponding eigenvalue-eigenvector pair) can be used to capture the spatiotemporal characteristics of the process. Specifically, each eigenvector represents an independent pattern of co-active brain regions and its corresponding eigenvalue describes the oscillation frequency of the activation pattern, see details in S3 Section in S1 File.

An example of these is provided in both Fig 9, where the larger the value in a given region defined by an arbitrary unit (AU), larger is the involvement of that region in the underlying dynamics at any given time. Additionally, at the different regions, different dynamical signals exhibit a variety of behaviors mainly captured by the stability and frequency in the vertical and horizontal axis of the central plot, respectively. Specifically, the lower the stability, the dynamical signal will vanish in a short period of time, whereas if it gets closer to one, then it will oscillate continuously. On the other hand, the frequency dictates how fast or slow the signal varies between its peaks. Altogether, it is apparent that a combination of regions has a superposition of the dynamical activities that are captured by the combination of stability and frequency.

To capture these spatiotemporal dynamics of the system, we use our method to filter the BOLD time series of each of the 98 subjects in all 4 runs in the HCP dataset. Further, eigenvectors from all the subjects before and after filtering are computed and clustered into $k = 5$ clusters using $k$-means clustering [45] to capture the resting state networks [46]. For in-depth analysis of the eigenvector clusters' community organization and the selected resolution, see [47]. Fig 9 shows the 5 clusters with associated eigenvectors of the resting-state BOLD signals in the left panel and the corresponding ARFIMA (1, $d$, 0) filtered BOLD signals in the right panel, plotted on the brain overlays (also known as "eigen brains"). In the figure, the nomenclature used in this study is as follows: the eigen brains inside blue colored box correspond to cluster 1, cluster 2 is outlined in orange colored box, cluster 3 in yellow outline, cluster 4 in purple colored box and cluster 5 in green colored outlined box.

The spatial correlation shown in Fig 10 between the identified clusters (through clustering of eigenvectors before and after filtering) and the seven resting-state networks (RSNs): visual (Vis) network, somatomotor (SM) network, dorsal attention network (DN), ventral attention network (VN), limbic network, executive control network (ECN) and default mode network (DMN) identified in [48] reveals that each cluster consists of one or more RSNs and the contribution of each RSN in each cluster (before and after filtering) remains the same. Furthermore, S1 Table compares the $R^2$ statistic and $p$-value of the spatial correlation between cluster

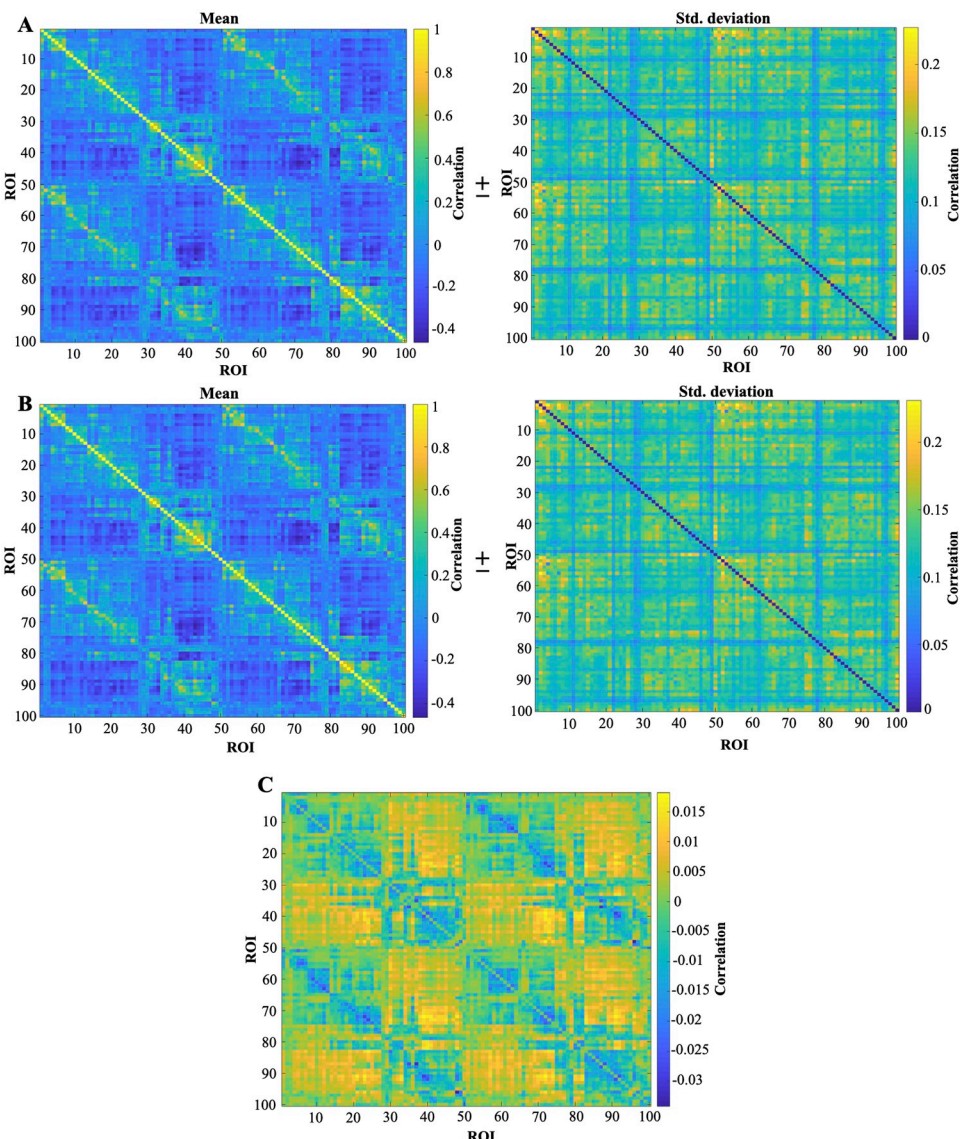

**Fig 7. Whole brain FC matrix (comprising of 100 ROI) defined based on Pearson's correlation.** The Pearson's correlation FC matrix of each subject in each run is averaged across to find one representative FC matrix (mean ± std. deviation). (A) Pearson's correlation matrix (mean ± std. deviation) (for 100 ROI) obtained from the resting-state BOLD time series. (B) Pearson's correlation matrix (mean ± std. deviation) (for 100 ROI) obtained from the ARFIMA (1, $d$, 0) filtered BOLD time series. (C) The difference between the mean Pearson's correlation matrix of the resting-state BOLD dataset(A) and the filtered BOLD time series of the whole brain (B).

centroid and RSNs before and after filtering. Visual comparison of the so-formed 5 clusters in Fig 9 reveal that cluster 1 to 4 (in blue, orange, yellow and purple boxes) looks identical. However, some changes in the visual and default mode RSNs in cluster 5 (green box) can be observed.

Additionally, the two-sample Kolmogorov–Smirnov test [44] was used to test the statistical similarity between the eigenvector cluster centroid of the respective cluster before and after clustering at a significance level of 0.05. The test results failed to reject the null hypothesis that the corresponding eigenvector cluster centroid before and after filtering are significantly similar.

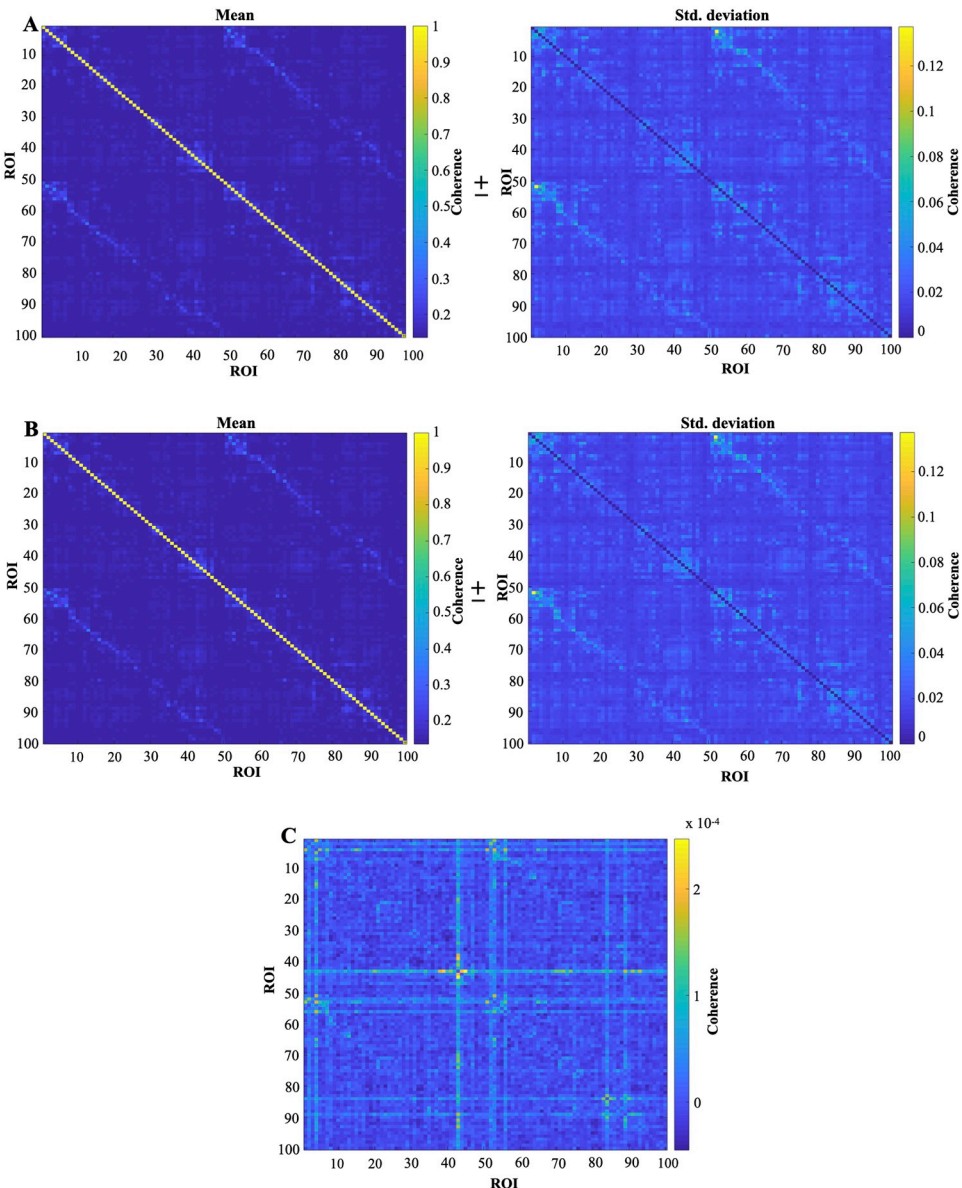

**Fig 8. Whole brain FC matrix (comprising of 100 ROI) based on the coherence.** The coherence FC matrix of each subject in each run is averaged across to find one representative FC matrix (mean ± std. deviation). (A) Coherence FC matrix (mean ± std. deviation) (for 100 ROI) obtained from the resting-state BOLD time series. (B) Coherence FC matrix (mean ± std. deviation) (for 100 ROI) obtained from the ARFIMA (1, $d$, 0) BOLD time series. (C) The difference between the mean coherence FC matrix of the resting-state BOLD dataset (A) and the filtered BOLD time series of the whole brain (B).

Nonetheless, the result of the implementation of the proposed filter on the temporal dynamics of the system is observed by witnessing the variation in the spectral content given by the associated eigenvalues of the resting-state and ARFIMA filtered BOLD signals. The magnitude and argument of the eigenvalue describe the stability and frequency respectively of the signals undergoing in the regions indicated by the associated eigenvectors. The top and the bottom central plot of Fig 9 shows the distribution of the eigenvalues (frequency vs stability) before and after ARFIMA (1, $d$, 0) filtering, respectively.

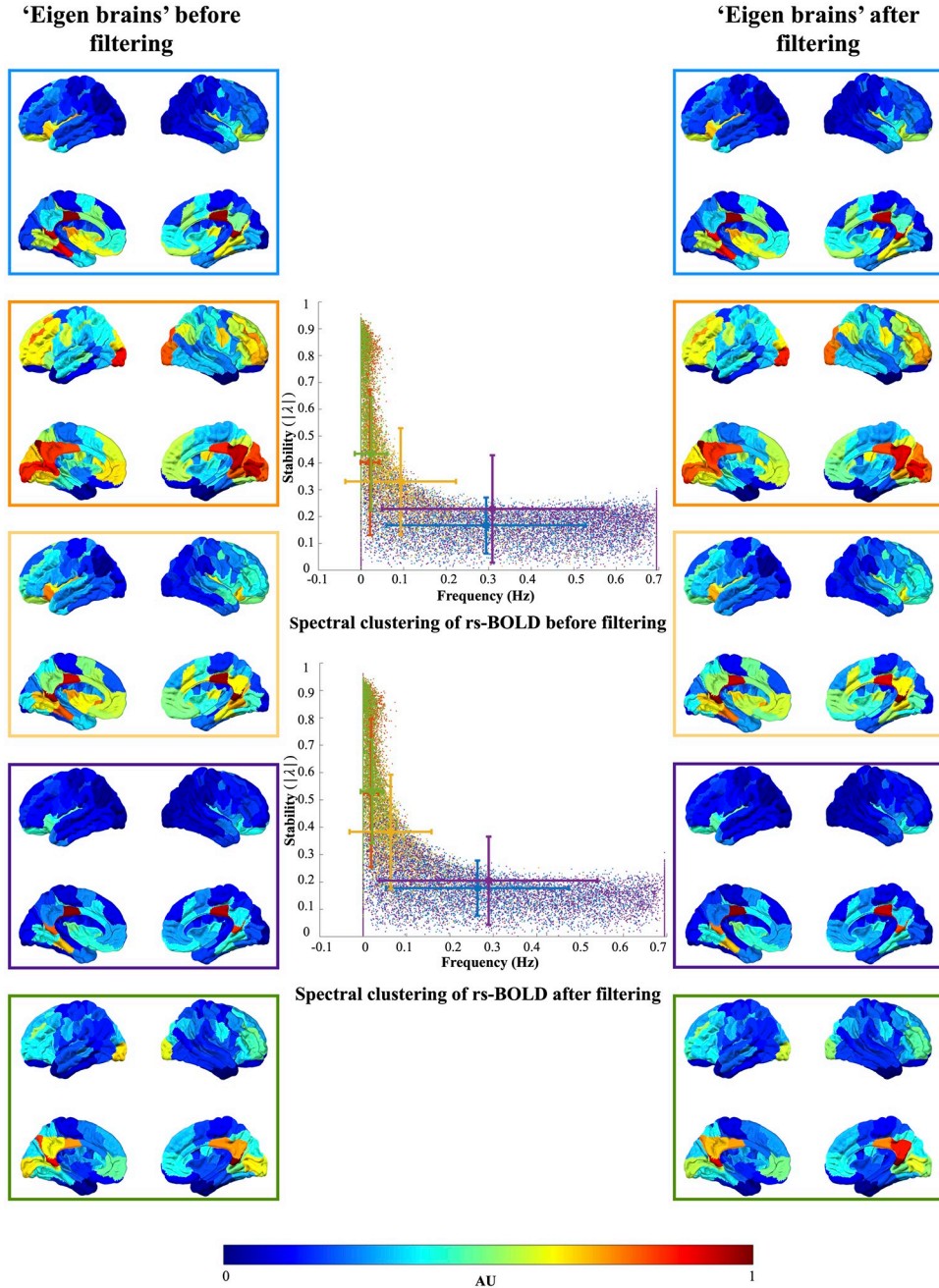

**Fig 9. Clustering of eigenvectors and eigenvalues obtained from resting-state BOLD signal and ARFIMA (1,** *d***, 0) (model-based) filtered resting-state BOLD signals.** All eigenvectors from all subjects were normalised and clustered into 5 clusters using *k*-means clustering. The clusters were color-coded across all subjects and all runs (98 subjects × 4 runs × 100 eigenmodes = 39,200 eigenvalues). The color codes blue, orange, yellow, purple and green correspond to cluster 1, 2, 3, 4 and 5, respectively. The plot in the centre shows the distribution of eigenvalues based on their frequency (the argument of eigenvalue) and stability (the absolute magnitude of eigenvalue) of resting-state and ARFIMA filtered BOLD signals. Error bars represent the mean and standard deviation of the average eigenvalue of each cluster. The eigenvalues are color coded based on the five identified clusters. The brain overlays in the left and right panel represent the spatial distribution of the eigenvector corresponding to the eigenvalue (same color coded) of resting-state BOLD signals before and after filtering, respectively. The cluster centroid (plotted on the brain overlays) were normalised by subtracting each centroid by its minimum element. Colorbar represents the normalised values of cluster centroid for each cluster (left and right panel).

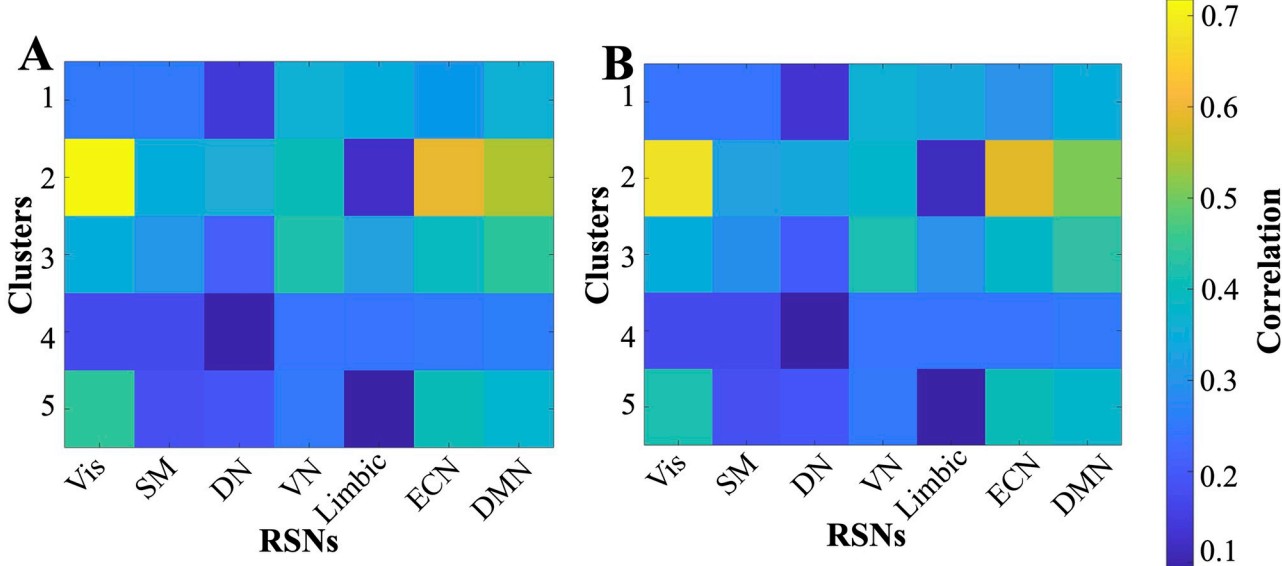

**Fig 10. Similarity between eigenvector centroid of the clusters and RSNs.** (A-B) denotes the spatial correlation between clusters and resting state networks before and after ARFIMA filtering on the resting-state BOLD data, respectively. The colorbar depicts the correlation values for both panels.

Table 2 compares the average value of the stability and frequency between each cluster of the eigenvalues of the preprocessed system and ARFIMA filtered system. The comparison shows that the stability of the system improves by around 7% in cluster 1, around 25% in cluster 2 and 5, and by approximately 16% in cluster 3 after the implementation of proposed filtering. To test the hypothesis that the distribution of frequency and stability is significantly different before and after filtering, Wilcoxon rank sum test [49] is utilised. The Wilcoxon rank sum test ($p < 0.05$) was performed between the respective clusters before and after filtering. The results of the statistical test revealed that their distribution was statistically different for all the clusters.

## Discussion

The long-term memory in the resting-state BOLD signals have been identified through fractal modelling (self-similarity structures) in various fMRI studies [39, 40, 50]. Herman *et al.* in [51] studied fractional properties in spontaneous BOLD fluctuations of a rat brain. Wang *et al.* used them to study the effect of different levels of isoflurane anaesthesia on the BOLD fluctuations [8]. The long-memory processes in temporal domain are modelled using ARFIMA (fractional) models [52]; however, to the best of our knowledge they have not been utilised in the context of filtering BOLD signals.

**Table 2. Comparison of the mean stability (magnitude of eigenvalue) and mean frequency (argument of eigenvalue) of the clusters before and after filtering.**

|  | Mean frequency (mean ± std. deviation) | | Mean stability (mean ± std. deviation) | |
|---|---|---|---|---|
|  | Before filtering | After filtering | Before filtering | After filtering |
| Cluster 1 | 0.2943 ± 0.2341 | 0.2637 ± 0.2112 | 0.1665 ± 0.1037 | 0.1773 ± 0.1011 |
| Cluster 2 | 0.0210 ± 0.0219 | 0.0185 ± 0.0167 | 0.4015 ± 0.2713 | 0.5250 ± 0.2696 |
| Cluster 3 | 0.0930 ± 0.1300 | 0.0632 ± 0.0948 | 0.3304 ± 0.1981 | 0.3832 ± 0.2093 |
| Cluster 4 | 0.3090 ± 0.2598 | 0.2890 ± 0.2509 | 0.2281 ± 0.1997 | 0.2044 ± 0.1617 |
| Cluster 5 | 0.0236 ± 0.0383 | 0.0209 ± 0.0262 | 0.4340 ± 0.2904 | 0.5312 ± 0.1893 |

A major challenge for evaluating the denoising method of the resting-state BOLD signal is the unavailability of the ground-truth signal. Therefore, first, we evaluate the proposed filter with a principled synthetic BOLD signal which has statistical properties similar to the resting-state BOLD signal identified in the literature (i.e., long-term memory and low-frequency fluctuations [9, 39–43]). The proposed denoising procedure was capable of retrieving the ground-truth signal from the artificial noise-induced synthetic BOLD signal. Thus, it provides evidence that the filter is attaining its objective.

It is worthwhile noticing that the approach for the estimation of fractional difference parameter *d* was a grid search approach such that the fractional differencing achieves short-term memory and yields stationary time series. The order of autoregressive component was limited to 1 to reduce the degrees of freedom and to obtain a stable model as their weights serve as a scaling factor of the differentiated BOLD. Additionally, we furloughed the moving average component to ensure that we deal with causal filters.

## Impact on power spectrum

Subsequently, we implemented the same methodology with the resting-state BOLD signals. The comparison of the power spectrum of the resting-state BOLD signals before and after filtering often leads to changes that seem to affect mainly the high-frequency components. For instance, see Fig 5, where we present three ROIs having different characteristics in their power spectrum. Fig 5A depicts the effect of filtering on the normalised power spectrum of an ROI which has most of the power at lower frequencies and a little component of power at higher frequencies. In contrast, Fig 5B shows the effect of filtering on the normalised power spectrum of an ROI with power spread out over the whole of its frequency range. Finally, Fig 5C corresponds to the ROI which has most of the power in its lower frequencies but still consists of the significant power in the higher frequency region. Despite depicting the behaviour similar to a low-pass filter (LPF), we notice the characteristics of LPF (first-order butterworth) and the proposed ARFIMA filter are different. It can be observed from Fig 11A that the LPF rejects the higher frequencies, but the derived filter does not eliminate them completely rather, it attenuates the high-frequency component while keeping some of the information that might be relevant to the neural activity. The magnitude bode plot (Fig 11B) of the transfer function of both the filters explains the observed behaviour. The attenuation is of the order of 50 dB at higher frequencies in case of low-pass Butterworth filter and of the order of 6 dB with the derived ARFIMA (1, 0.3, 0) filter. Hence, it provides evidence that the derived filter has different characteristics from the ordinary LPF.

Furthermore, because the ARFIMA model-based approach is done in the time-domain, a variety of other scenarios are expected. Specifically, for most signals, ARFIMA filtering performs attenuation at higher frequencies as can be seen in Fig 5, but we found sensitivity in some frequency distributions where, a small amount of amplification was observed in the higher frequencies. For instance, Fig 12A shows one such ROI where the high-frequency components are actually amplified. The amount of amplification can be observed in the magnitude bode plot of the proposed filter in Fig 12B, which is of the order of 0.1 dB. This provides evidence of the versatility of the proposed filtering scheme which is tailored for each of the channels and the associated data under consideration. Although we lack the ground-truth of these signals, the synthetic examples explored provide converging evidence that the proposed filters are attaining their objective.

The primitive studies denote that the fluctuations due to neural activity in resting-state BOLD signals are associated with the low-frequencies [1, 53, 54], however, technical advances in the MR imaging techniques led to the identification of RSNs in frequencies higher than

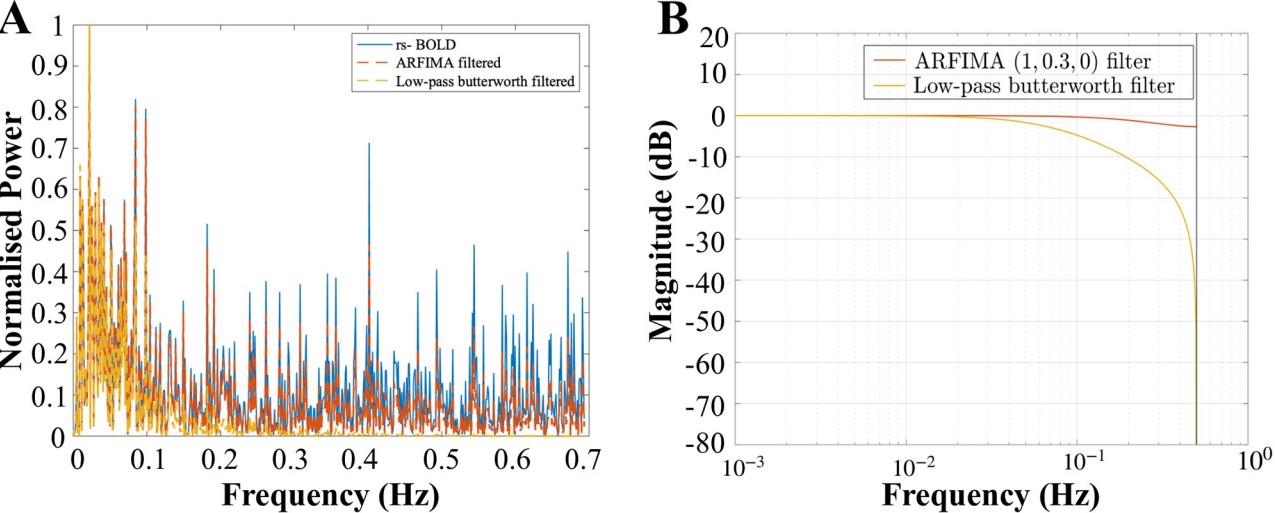

**Fig 11. Comparison between the derived ARFIMA filter and first-order low-pass butterworth filter.** Both the figures show the comparison made on ROI: 11 corresponding to the somatomotor auditory region of the brain. (A) Normalised power spectrum of the unfiltered and filtered BOLD signal. The normalised power spectrum corresponding to the preprocessed resting-state BOLD signal is represented by the cyan curve, low-pass first-order butterworth filtered by yellow curve and the ARFIMA $(1, 0.3, 0)$ filtered BOLD by the orange curve. (B) Magnitude bode plot of the transfer function of the low-pass first-order butterworth filter (yellow curve, cut off frequency: 0.1Hz) and the derived ARFIMA $(1, 0.3, 0)$ filter (orange curve).

0.1Hz [55–57]. The latter suggests that the high-frequency fluctuations in resting-state BOLD signals are not only the result of artefacts but also include contribution from underlying neuronal activity. Therefore, completely filtering out the high-frequency components do not seem to be an ideal denoising approach. Thus, it is remarkable to highlight that the proposed filter seems to be able to attenuate/amplify the higher frequencies when required.

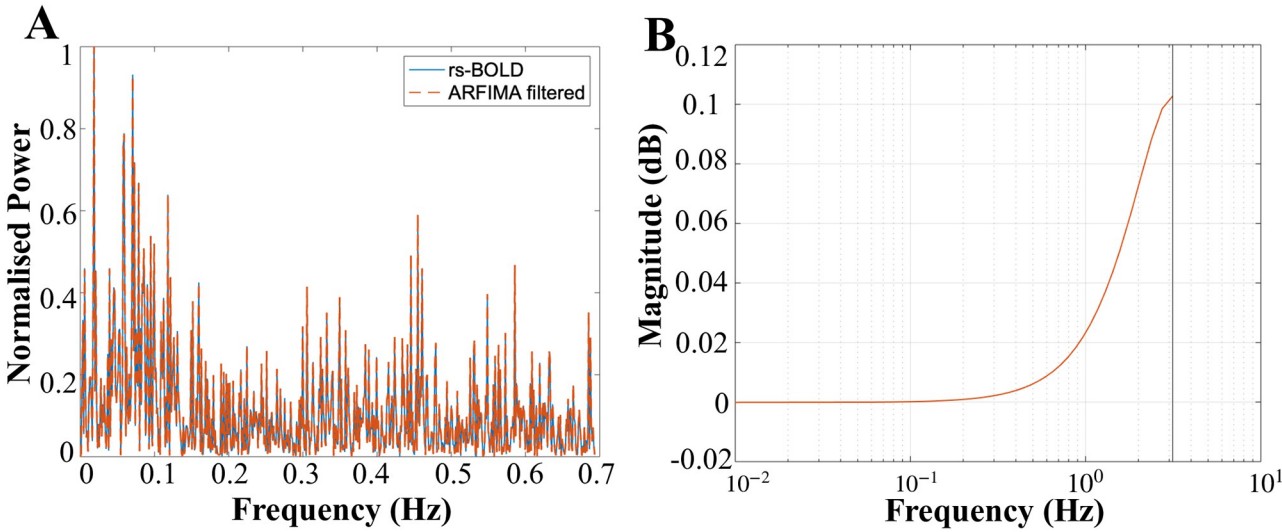

**Fig 12. Amplification in the high frequencies for one of the ROI.** (A) shows the normalised power spectrum of the resting-state BOLD signal (cyan curve) and the ARFIMA filtered signal (dashed orange curve) for one of the ROI. (B) Magnitude bode plot of the derived ARFIMA filter (for the ROI shown in (A)) depicting amplification in higher frequencies.

## Impact on FC measures

Next we look into functional connectivity matrices between different ROIs of brain, particularly, Pearson's correlation and coherence. Since, FC is dominated by the high power in slow frequencies; therefore, these measures capture the slow-frequency properties of the signal. As we observe from the power spectrum analysis that the effect of filtering seems to be more heavily present at the high-frequency component of the signal, hence, the FC matrix may not observe any significant changes due to filtering. Figs 7 and 8 indeed depicts that the mean FC matrices before and after filtering are similar. Additionally, the two-sample Kolmogorov–Smirnov test [44] on pre and post-filtered mean FC matrices failed to reject the null hypothesis that they are statistically similar (at a significance level of 0.05).

## Impact on eigenmode analysis of directed connectivity

Since, FC measures were not sensitive enough to reveal what is changed by filtering, therefore, we perform subject-level eigenmode decomposition and *k*-means clustering to: *a)* reveal the heterogeneous spatial profile of the eigenmodes of the brain and *b)* to leverage these now teased out clusters to better localize any changes after filtering (that potentially impacted some brain networks differently [possibly due to their intrinsic spectral profile]).

Finally, we showed that this detailed analysis allowed us to highlight filtering-related changes. More intriguingly, the spectral profiles associated with the clusters changes after filtering. The Wilcoxon rank sum test [49] on the corresponding distribution of the eigenvalues of each cluster before and after filtering indicates that they are statistically different ($p < 0.05$). Thus, providing additional evidence that the proposed filter modifies the (spatiotemporal) spectral content of the resting-state BOLD signals. These suggest that the proposed methodology filtered out the data that was not consistent with the proposed models which evidence suggests to be properly modeled by fractional-order processes and, further, corroborated by the synthetic examples explored. That said, we conjecture that we were able to remove some additional noise that could be due to sporadic activity not eliminated through data preprocessing, or maybe due to additional artefacts introduced by such preprocessing.

## Methodological considerations and limitations

A major challenge in the evaluation of fMRI denoising methods is the unavailability of the standard signal for distinguishing fluctuations due to neural component from artefacts. Therefore, it is impossible to state beyond doubt that a signal being filtered out is the noisy component or was the source of neural activity.

It is important to highlight that the structured recording noise, such as autocorrelated noise, has the potential to negatively impact the modeled system [58, 59]. Although we have regressed out the global mean signal [60] to account for the shared global noise [61–63], our model is unable to account for other unknown structured (e.g., autocorrelated) and time-varying recording noise [58, 59]. Furthermore, artifacts may be introduced by global mean signal regression (GSR), since it removes any global activation patterns (e.g., vigilance [64] or arousal [65]) in addition to the shared noise and can change the correlation structure. This noise reduction method's limits are the focus of ongoing debate [66]. We decided to use this preprocessing step after weighing the possible disadvantages of GSR against the primary concerns about substantial global artifacts like cardiac and respiratory noise. Nonetheless, it would be useful to investigate the global signal's spectral profile and the effect of GSR on the modeled system's spectral properties.

It is worth noting that some of the preprocessing steps, for instance removal of structured and unstructured noise can impact the reported results. Although beyond the scope of the

current proof-of-concept work, future work should explore how different preprocessing steps such as mentioned by Glasser *et al.* in [67], the data-driven ICA-based noise removal, can effect the ARFIMA model, and consequently filtered outcomes.

Nonetheless, we created a synthetic BOLD signal which follows the observed properties of the original BOLD signals and consists of low-frequency fluctuations. Furthermore, there are some methodological limitations. For instance, one of the main parameter in the ARFIMA filtering is the fractional difference parameter, *d*. Selecting a value too high, may lead to over differencing of the time series and thus, can introduce artificial memory [21, 32]. Liu *et al.* in [28] illustrate that the estimation of *d* with different methods or software can result in different values. This is because each of these methods seeks to minimise different objective functions tailored to their requirements [68]. Therefore, a novel method can be developed to estimate *d*, which can also help improve the results obtained in this study.

Additionally, the current research focusses on the univariate filtering of the resting-state BOLD signals, however, the spatiotemporal dependencies between different ROIs of a brain cannot be ignored [69, 70]. Thus, future work should focus on extending the proposed filter to the multivariate domain. Additionally, the value of fractional differencing parameter *d* in this study was considered constant over a period of time for resting-state BOLD time series for a particular ROI. Future research should account for possible variations, as those that occur in criticality analysis of electrocorticogram signals in epilepsy [71] patients, or critical transition phenomena found in nature [72].

The proposed ARFIMA filter is limited to the field of resting-state BOLD signals for this research. Nonetheless, its scope can be extended to various signals which exhibit long-memory property such as financial time series, electroencephalography signal and underwater signal [28] and in numerous fields, namely, signal processing, control engineering, biomedical systems and physics (Magin *et al.* in [73] provides a reference to literature in each of these domains). As noticed in this study, the proposed filters are suitable for modelling the low-frequencies, therefore, their usage can be advantageous where a low-frequency signal is of paramount interest.

Finally, we have looked into static FC as one of the metric, largely based in their common usage in the resting-state fMRI data analysis [74, 75]. However, recently FC has been observed to fluctuate over time [76] and has opened window to the new research area called dynamic functional connectivity (dFC) [77, 78]. Future work could observe the effect of the proposed filter on the dFC metrics.

## Supporting information

**S1 File. Suppoting information.** In this supplementary information file, we provide complementary information that aims to sharpen the intuition behind the usage of different matrices for analysing the effect of proposed filter on resting state BOLD signals.
(PDF)

**S1 Fig.**
(TIFF)

**S1 Table. $R^2$ statistic and p-value of the spatial correlation between the cluster centroid and the identified RSNs.** The comparison of these statistics show that the presence of RSNs in each of the identified clusters before and after proposed filtering is similar.
(PDF)

## Author Contributions

**Conceptualization:** Ishita Rai Bansal, Sérgio Pequito.

**Data curation:** Maxwell Bertolero, Danielle S. Bassett.

**Formal analysis:** Ishita Rai Bansal, Arian Ashourvan.

**Investigation:** Ishita Rai Bansal.

**Methodology:** Ishita Rai Bansal, Sérgio Pequito.

**Resources:** Ishita Rai Bansal, Danielle S. Bassett, Sérgio Pequito.

**Software:** Ishita Rai Bansal.

**Validation:** Ishita Rai Bansal, Arian Ashourvan.

**Visualization:** Ishita Rai Bansal, Arian Ashourvan.

**Writing – original draft:** Ishita Rai Bansal.

**Writing – review & editing:** Arian Ashourvan, Maxwell Bertolero, Danielle S. Bassett, Sérgio Pequito.

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
