## [Decision Letter · Decision Letter 0]

27 Jan 2022

PONE-D-21-22658Model-based stationarity filtering of long-term memory data applied to resting-state blood-oxygen-level-dependent signalPLOS ONE

Dear Dr. Pequito,

Thank you for submitting your manuscript to PLOS ONE. After careful consideration, we feel that it has merit but does not fully meet PLOS ONE’s publication criteria as it currently stands. Therefore, we invite you to submit a revised version of the manuscript that addresses the points raised during the review process.

I received the reports of three referees, whose comments are appended below. As you will see, the referees were supportive, but they raised important points and made suggestions for modifications, which they felt must be addressed before this work would be acceptable for publication

Based on this recommendation, I therefore invite you to revise and resubmit your manuscript, taking into account all the points raised.

Best regards,

Federico Giove, Academic Editor

We look forward to receiving your revised manuscript.

Kind regards,

Federico Giove, PhD

Academic Editor

PLOS ONE

Journal Requirements:

2. Please note that in order to use the direct billing option the corresponding author must be affiliated with the chosen institute. Please either amend your manuscript to change the affiliation or corresponding author, or email us at plosone@plos.org with a request to remove this option.

Reviewers' comments:

Reviewer's Responses to Questions

**Comments to the Author**

1. Is the manuscript technically sound, and do the data support the conclusions?

Reviewer #1: Yes

Reviewer #2: Yes

Reviewer #3: Partly

2. Has the statistical analysis been performed appropriately and rigorously? 

Reviewer #1: Yes

Reviewer #2: Yes

Reviewer #3: I Don't Know

3. Have the authors made all data underlying the findings in their manuscript fully available?

Reviewer #1: Yes

Reviewer #2: Yes

Reviewer #3: Yes

4. Is the manuscript presented in an intelligible fashion and written in standard English?

Reviewer #1: Yes

Reviewer #2: Yes

Reviewer #3: Yes

5. Review Comments to the Author

Reviewer #1: In “Model-based stationarity filtering of long-term memory data applied to resting-state

blood-oxygen-level-dependent signal", Bansal et al. propose a new method to filter unknown source of noise present in resting-state blood-oxygen-level-dependent (BOLD) signal acquires through functional MRI. The proposed method is based on autoregressive fractional integrative moving average (ARFIMA) process. The study is very interesting, well written, and well reported. However, there are certain confusions regarding different parameters’ choices, which need clarification by the authors.

Specific comments to the authors are given below.

1. Materials and Methods:

a. Why are resting-state fMRI (rs-fMRI) time series assumed to be stationary in the study? Based on previous studies resting-state fMRI time series are non-stationary due to presence of various artifacts (Piaggi1, Menicucci et al. 2011, Patel, Kundu et al. 2014, Guan, Jiang et al. 2020, Davey, Grayden et al. 2021). Authors should provide references about the rs-fMRI time series being stationary.

b. What can be the possible advantage of using fractional differencing (in ARFIMA) instead of just differencing (in ARIMA) for filtering? ARIMA may be simpler to implement compared to ARFIMA.

c. The choice of ARFIMA process is used for both stationary & non-stationary time series based on the values of the fractional difference ‘d’ (-0.5<d<0.5 and="" d="" for="" series="" stationary="" time="">0.5 for non-stationary case) (Lopes, Olbermann et al. 2004, Box, Jenkins et al. 2016). However, the value of ‘d’ is estimated from the interval [0.1, 5.0] and the estimated values of ‘d’ are outside the interval [-0.5, 0.5] for both synthetic and BOLD fMRI time series. The authors should provide the reason of ignoring this difference.

d. It is suggested that authors introduce non-stationarity in the synthetic time series and perform filtering on both stationary and non-stationary synthetic time series using ARFIMA process.

e. It is mentioned that differencing is used to remove non-stationarity in the time series (page 4), however, in the study it is applied on rs-fMRI assuming it to be stationary. Authors should comment on this.

2. Results:

a. Why is the frequency range for the synthetic data restricted to 0.1-0.15 Hz? This is the frequency range obtained after the low-pass filtering of the rs-fMRI time series and the study is targeting to look into the frequencies above this range. Keeping this in mind the synthetic data should have higher frequencies included in it, too.

b. What conclusion can be drawn from the result that almost 50% of normalized power spectrum differences (before and after filtering) are not significant?

c. Why eigenvectors (from eigenmode analysis) are clustered in five clusters?

3. Discussion:

a. The results report attenuation in high frequencies after filtering with ARFIAM process and amplification in few cases. What are the relative percentages of attenuations and amplifications? Can something be concluded about the reason of attenuation or amplification? What does it mean by’ the proposed filter seems to attenuate/amplify the higher frequencies, when required? It is not clear ‘required’ by what?

Minor points

1. The form of the fractional differencing mentioned in (4) is specifically for the case d>-1 (Xiu and Jin 2007), which is not mentioned specifically in the study.

2. Caption of the Figure 5 is inverted.

3. Figure 9 is inverted.

References:

Box, G., G. Jenkins, G. Reinsel and G. Ljung (2016). Time Series Analysis Forecasting and Control (Fifth Edition).

Davey, D., D. Grayden and L. Johnston (2021). "Correcting for Non-stationarity in BOLD-fMRI Connectivity Analyses." Front Neurosci.

Guan, S., R. Jiang, H. Bian, J. Yuan, P. Xu, C. Meng and B. Biswal (2020). "The Profiles of Non-stationarity and Non-linearity in the Time Series of Resting-State Brain Networks." Front Neurosci.

Lopes, S., B. Olbermann and V. Reisen (2004). "A COMPARISON OF ESTIMATION METHODS IN NON-STATIONARY ARFIMA PROCESSES." Journal of Statistical Computation and Simulation.

Patel, A., P. Kundu, M. Rubinov, P. Jones, P. Vértes, K. Ersche, J. Suckling and E. Bullmore (2014). "A wavelet method for modeling and despiking motion artifacts from resting-state fMRI time series." NeuroImage.

Piaggi1, P., D. Menicucci, C. Gentili, G. Handjaras, M. Laurino, A. Piarulli, M. Guazzelli, A. Gemignani and A. Landi (2011). Adaptive Filtering for Removing Nonstationary Physiological Noise from Resting State fMRI BOLD Signals International Conference on Intelligent Systems Design and Applications

Xiu, J. and Y. Jin (2007). "Empirical study of ARFIMA model based on fractional differencing." Physica A.</d<0.5>

Reviewer #2: This manuscript describes the unique application of the autoregressive fractional integral moving average (ARFIMA) filter to resting-state BOLD synthetic and real data. The manuscript reports that ARFIMA can attenuate higher frequencies without impacting on traditional FC measures. Furthermore, they demonstrate filtered data has improved “stability” as estimated using eigenmode analysis of directed connectivity.

This method has merit for filtering out true “noise” from the fMRI and would be more nuanced alternative to traditional spectral filters which are often used in rsfMRI pre-processing. However, I think the clarity of the manuscript undermines the import of the topic.

1. It is not explicitly stated, but I *think* that the rationale for the ARFIMA filter in this context is that it will retain signal with a long-term autocorrelation structure presumed to be neural signal, and filter out noise without this autocorrelation structure. This seems like a reasonable motivation to me; however, I feel that the manuscript lacks a clear and explicit exposition of this motivation. It is attempted in the second-to-last paragraph of the introduction but it could be a lot clearer and more explicit. On first read, it feels like the motivation for ARFIMA was largely that it has never been using in this context before…

2. The term “denoising” is often loosely used in the rsfMRI literature to refer to the removal of both unwanted structured artefacts and/or unstructured “noise” in the more traditional sense. The authors seek to clarify that their proposed method deals only the latter (filtering unstructured noise), and not structured artefact, nonetheless this needs to be emphasised as it has the potential to be missed by the reader which will result in confusion for readers that are used to the more liberal interpretation.

3. You have chosen to include GSR within the definition of “minimally” pre-processed. This is a controversial choice that not all in the community would agree with. I don’t think the inclusion or exclusion of the global signal, which is a structured signal, would impact the performance of ARFIMA, but this should be minimally addressed, and ideally even demonstrated…

4. ICA-FIX seeks to remove both structured and unstructured noise from BOLD therefore ARFIMA is being presented with BOLD data that already has attenuated unstructured noise. It would be interesting to use ICA to remove only the structured noise, and then see how well ARFIMA performs on the unstructured noise (without the prior attenuation). It seems plausible to me that ARFIMA may do better on unstructured noise than ICA-FIX and could/should potentially be used in conjunction with FIX.

5. Figure 3 presents metrics for the synthetic BOLD. It would be valuable to the see the same metrics for the pre-processed bold alongside the synthetic. Furthermore, I would prefer to also see an example time-course from both the BOLD and synthetic data.

6. The non-symmetric colorbars in figures 7 and 8 make it harder to interpret the matrices.

7. Table 2 would be easier to interpret as a bar plot.

8. I found it odd that new figures were introduced in the discussion.

Reviewer #3: The authors conduct a potentially interesting fMRI filtering study that aims to address the problem of unstructured noise. Unfortunately, the manuscript is let down by issues of presentation and I found the methods confusing. I list several analyses that would need to be performed for evaluation before I would recommend publication.

Major Concerns:

1) Manuscript organization for Peer Review: Today most manuscripts are reviewed on a computer as an electronic PDF file. As a result, it is very problematic to put the figures at the end of the document, as this requires the overworked peer reviewer to scroll back and forth in the document. Figures and legends should be embedded within the manuscript at logical locations and high-quality PDF conversion for figures should be handled by the authors (i.e., don’t rely on poorly configured journal/publisher websites) to ensure that the manuscript is presentable to the peer reviewers. This allows the reviewer to focus on your science and not problems with the organization. I have scrolled through the document once, but may have missed things.

2) It appears that the authors are starting from the data that has not been cleaned by spatial ICA+FIX. This is incorrect. If they are interested in working on the unstructured noise problem, they should work with data containing only neural signal and unstructured noise. Later it looks like spatial ICA+FIX was in fact used. I would not use the terminology, “minimal preprocessing,” as most folks take that to mean the outputs of the HCP’s spatial preprocessing pipelines (i.e., Glasser et al., 2013 Neuroimage), before temporal preprocessing (like linear trend removal or sICA+FIX).

3) What is the justification for using global signal regression in this study? Glasser et al., 2018; 2019 Neuroimage shows that the global signal is a roughly 50/50 mix of neural signal and global respiratory artifact.

4) It is said that the “98 subjects with the least movement artifacts” were used. Why? sICA+FIX should have already addressed the spatially specific effects of head motion.

5) I must admit I did not follow the filtering procedure at all. Probably what the authors need to show to convince me that it is not doing something bad is all of the following: 1) Show the effect of the filtering on i) The signal ICA component timeseries and ii) The unstructured noise timeseries after performing the filtering on the overall timeseries. This could be accomplished by taking the filtered timeseries and spatially regressing the signal sICA component spatial maps into the filtered timeseries to generate filtered component timeseries. These could then be compared with the originals to ensure that the filtration approach is not affecting the neural signal. Also, the overall unstructured noise variance after filtration should be compared to that before filtration. The unstructured noise variance can be obtained by regressing out the signal ICA components and taking the variance of the residuals. 2) Show the difference in task fMRI activation betas before and after this filtration procedure. The betas should not change much and ideally the pattern of change should match the overall activation pattern, but the zstats might improve (though they should be corrected using mixture modelling, such as that available in melodic, before and after filtration).

Minor Comments:

1) Line 18: Marcus et al had errors in the formulas. Use Glasser et al., 2018 Neuroimage Supplemental topic #3 for more accurate measures.

2) The spatial preprocessing of Glasser et al., 2013 Neuroimage should be mentioned.

3) It is not said whether volume or CIFTI grayordinates data is used here. CIFTI is recommended.

6. PLOS authors have the option to publish the peer review history of their article (what does this mean?). If published, this will include your full peer review and any attached files.

Reviewer #1: **Yes: **Sadia Shakil

Reviewer #2: No

Reviewer #3: No

---

## [Author Response · Author response to Decision Letter 0]

5 Apr 2022

Find rebuttal letter enclosed in the attached pdf.

---

## [Decision Letter · Decision Letter 1]

9 May 2022

Model-based stationarity filtering of long-term memory data applied to resting-state blood-oxygen-level-dependent signal

PONE-D-21-22658R1

Dear Dr. Pequito,

We’re pleased to inform you that your manuscript has been judged scientifically suitable for publication and will be formally accepted for publication once it meets all outstanding technical requirements.

Kind regards,

Federico Giove, PhD

Academic Editor

PLOS ONE

Additional Editor Comments (optional):

Reviewers' comments:

Reviewer's Responses to Questions

**Comments to the Author**

1. If the authors have adequately addressed your comments raised in a previous round of review and you feel that this manuscript is now acceptable for publication, you may indicate that here to bypass the “Comments to the Author” section, enter your conflict of interest statement in the “Confidential to Editor” section, and submit your "Accept" recommendation.

Reviewer #1: All comments have been addressed

Reviewer #2: All comments have been addressed

Reviewer #3: (No Response)

2. Is the manuscript technically sound, and do the data support the conclusions?

Reviewer #1: Yes

Reviewer #2: Yes

Reviewer #3: (No Response)

3. Has the statistical analysis been performed appropriately and rigorously? 

Reviewer #1: Yes

Reviewer #2: Yes

Reviewer #3: (No Response)

4. Have the authors made all data underlying the findings in their manuscript fully available?

Reviewer #1: Yes

Reviewer #2: Yes

Reviewer #3: (No Response)

5. Is the manuscript presented in an intelligible fashion and written in standard English?

Reviewer #1: Yes

Reviewer #2: Yes

Reviewer #3: (No Response)

6. Review Comments to the Author

Reviewer #1: All of my comments are satisfactorily addressed. I believe the manuscript is now in the form to published.

Reviewer #2: (No Response)

Reviewer #3: (No Response)

7. PLOS authors have the option to publish the peer review history of their article (what does this mean?). If published, this will include your full peer review and any attached files.

Reviewer #1: No

Reviewer #2: No

Reviewer #3: No

---

## [Editor Report · Acceptance letter]

19 May 2022

PONE-D-21-22658R1 

Model-based stationarity filtering of long-term memory data applied to resting-state blood-oxygen-level-dependent signal 

Dear Dr. Pequito:

I'm pleased to inform you that your manuscript has been deemed suitable for publication in PLOS ONE. Congratulations! Your manuscript is now with our production department. 

Kind regards, 

on behalf of

Dr. Federico Giove 

Academic Editor

PLOS ONE